# Antimicrobial Properties of Carboxymethyl Cellulose/Starch/N’N Methylenebisacrylamide Membranes Endowed by Ultrasound and Their Potential Application in Antimicrobial Packaging

**DOI:** 10.3390/polym16091282

**Published:** 2024-05-03

**Authors:** Youliang Cheng, Xinyi Cheng, Changqing Fang, Jing Chen, Xin Zhang, Changxue Cao, Jinpeng Wang

**Affiliations:** Faculty of Printing, Packaging Engineering and Digital Media Technology, Xi’an University of Technology, Xi’an 710048, China; chengyouliang@xaut.edu.cn (Y.C.); cxy_13233291316@163.com (X.C.); chenjing@xaut.edu.cn (J.C.); zhangxinxut@163.com (X.Z.); changxue0110@163.com (C.C.); 18729187396@163.com (J.W.)

**Keywords:** sonochemistry, antibacterial properties, carboxymethyl cellulose

## Abstract

Cellulose is used widely in antimicrobial packaging due to its abundance in nature, biodegradability, renewability, non-toxicity, and low cost. However, how efficiently and rapidly it imparts high antimicrobial activity to cellulose-based packaging materials remains a challenge. In this work, Ag NPs were deposited on the surface of carboxymethyl cellulose/starch/N’N Methylenebisacrylamide film using ultrasonic radiation. Morphology and structure analysis of as-prepared films were conducted, and the antibacterial effects under different ultrasonic times and reductant contents were investigated. These results showed that Ag NPs were distributed uniformly on the film surface under an ultrasonic time of 45 min. The size of Ag NPs changes as the reducing agent content decreases. The composite film demonstrated a slightly better antibacterial effect against *E. coli* than against *S. aureus*. Therefore, this work can provide valuable insights for the research on antimicrobial packaging.

## 1. Introduction

Plastic packaging materials with poor degradability bring a series of environmental problems. Due to the depletion of nonrenewable resources and serious environmental pollution, replacing traditional synthetic plastic packaging with renewable, degradable, and environmentally friendly bio-based materials has become a popular trend in the packaging field [1]. Therefore, biological-based materials with renewable, biocompatible, and degradable properties have received widespread attention. Biodegradable materials include proteins (corn protein, whey protein, soy protein, etc.), polysaccharides (cellulose, starch), polylactic acid (PLA), polyvinyl alcohol (PVA), bacterial cellulose, and polyhydroxyalkylates (PHA). Among the materials mentioned above, cellulose has been an ideal material for multifunctional composite preparation due to its convenient water processing, good biodegradability, biocompatibility, natural abundance, and sustainability [2]. Cellulose-based materials are used widely in many fields, including environmental remediation [3], electronics, aerospace [4], catalytic degradation [5], food preservation [6], wastewater treatment [7], and antimicrobial materials [8,9]. 

Carboxymethyl cellulose has polyelectrolyte properties due to the presence of its weakly acidic carboxymethyl group, has the advantages of good water solubility, non-toxicity, and biocompatibility, and is widely used in seawater desalination [10], dye adsorption [11], and food industries [12], etc., but the mechanical properties of the film prepared by itself are quite poor. Starch is a kind of polymer carbohydrate, which has the advantages of good biodegradability, low price, and good film-forming properties. Therefore, a composite consisting of carboxymethyl cellulose and starch can prepare composite films with good mechanical properties; many studies have been conducted on the preparation of composite membranes by mixing carboxymethyl cellulose and starch. To further improve the plasticity and mechanical properties of composite films, plasticizers such as glycerin and sorbitol are usually introduced into the membrane-forming liquid to prepare these membranes [13].

During time spent in the supply chain, food can deteriorate and rot due to microbial activities, resulting in the wastage of food resources. At the same time, foodborne spoilage bacteria (such as *E. coli*, *S. aureus*, etc.) can reproduce in large quantities in these foods and produce toxins, which will enter the human body along with the food and cause foodborne illness, thus affecting the health of the people. Therefore, antimicrobial packaging materials can be prepared by introducing antimicrobial additives into cellulose-based materials that do not have antimicrobial activity by themselves, allowing them to be used more widely in the market. Silver nanoparticles (Ag NPs) are considered to be the most commonly used inorganic antibacterial material, which has been widely used in medicine [14], wound dressing [15], pharmaceutical [16], and other fields. Ag NPs have good antimicrobial performance and biocompatibility, which are suitable for incorporation into cellulose-based materials to prepare antimicrobial composites. In the case of Ag NP loading, Ag NPs were prepared and then dispersed in the matrix. The preparation of Ag NPs is divided into physical, chemical, and biological methods. The products of the physical method tend to agglomerate, the equipment technology requirement is high, the cost is high, and it is impossible to produce on a large scale. The products of the biological method are not high purity, and the reducibility is weak. The organic solvents used in chemical methods are hazardous to the human body and pollute the environment. Therefore, ultrasonic radiation-assisted in situ synthesis of nanocoatings are used to avoid toxic solvent use and make the coating process shorter, more efficient, and more eco-friendly, with more potential for antimicrobial composite preparation [17].

Sonochemical techniques for encapsulating nanoparticles in materials are based on a number of physical forces such as turbulence, microfluidics, microjets, and shockwaves that are generated by the collapse of cavitation bubbles in liquids when ultrasonic radiation is applied. When a bubble collapses, it generates a high-intensity shock wave that enhances the mass transfer effect, increasing the rate of chemical reactions and particle collisions that embed the newly formed nanoparticles in the material. The ultrasound can complement (enhance) traditional techniques with eco-friendly precursors and solvents. Ultrasound shows advantages in reactions involving material synthesis, such as shorter reaction times, higher yields and purity, and ambient/mild conditions. Bubbles generated by cavitation act as reactors and their asymmetric implosion leads to various physical effects of microfluidics, including high-speed microjets and high-intensity shock waves inducing effective mixing or stirring, enhancing localized heat and mass transfer; and reducing particle size and agglomeration, and changing particle morphology [18]. There are many reports on the preparation of antibacterial materials via sonochemistry methods, such as in the hygiene of fresh agricultural products [19]; preparation of edible films [20]; green synthesis of antibacterial metal oxides [21]; development of active nanomedicines [22]; dyeing of fiber materials [23]; research on antibacterial scaffolds [24]; food packaging films [25]; sterilization methods in the food industry [26]; wastewater treatment [27]; and improvement of marine environments [28]. Various substrates have been chosen during the sonochemistry process; however, it has been relatively rare to use cellulose-based materials as the packaging materials via sonochemical technology until now. Thus, it is an interesting issue due to the wide application of packaging materials.

In this work, carboxymethyl cellulose/starch/N’N Methylenebisacrylamide composite films were used as substrates, Ag NPs were synthesized in situ in one step on the surface of membranes via ultrasonic radiation, and the size of Ag NP particles was tuned by varying the ultrasound time and the content of the reducing agent. The morphology, structure, composition, and thermal stability of the composite films were characterized, and their antibacterial activity against *S. aureus* and *E. coli* was investigated. The composite films synthesized under sonochemical conditions exhibited good antibacterial activity.

## 2. Materials and Methods

### 2.1. Materials

Silver acid was provided by the China Shanghai, Shanghai Pharmaceutical Group Chemical Reagent Co., Ltd., carboxymethyl cellulose (CMC) was provided by the China Tianjin, Tianjin Fuchen Chemical Reagent Factory, ethylene glycol was provided by the China Tianjin, Tianjin Baishi Chemical Co., Ltd., and N’N methylene bisacrylamide (MBA) was provided by the China Tianjin, Tianjin Fuchen Chemical Reagent Factory. Starch and glycerol were provided by the China Tianjin, Tianli Chemical Reagent Co., Ltd. Agar nutrient was provided by the Aobaoxing Biotechnology Co., Ltd. in Beijing, China Beijing.

### 2.2. Preparation of Carboxymethyl Cellulose-Based Films

A total of 1 g of carboxymethyl cellulose, 1 g of soluble starch, and 1 g of MBA were dissolved in 49 mL, 49 mL, and 19 mL of deionized water, placed on a magnetic stirrer with a water bath temperature of 80 °C, heated, and stirred for 1 h to make them fully dissolved. The three solutions were mixed and stirred for 30 min to make the mixture homogeneous, and then 1 mL of glycerol was added and stirred a little to form a film solution. The liquid was poured into a glass petri dish and allowed to stand for a little time to remove the surface bubbles, and then dried in an oven at 60 °C to obtain the CMC/SR/MBA membrane (CSM).

### 2.3. Sonication Treatment of Carboxymethyl Cellulose-Based Films

A total of 0.2 g of AgNO_3_ was dissolved in 25 mL deionized water, then the precursor solution was obtained by adding ethylene glycol and placing it on a magnetic stirrer in a 70 °C water bath for 30 min. The above CSM film was cut into square membranes with a size of 1.5 cm × 1.5 cm and placed in the precursor solution for sonication treatment. Finally, the membrane was taken out, rinsed, and dried to obtain the Ag NPs/CMC/SR/MBA composite membrane (ACSM). The schematic diagram of the Ag NP coating on the surface of the ACSM composite membrane is shown in Figure 1.

Different composite membranes were obtained based on different reducing agent contents and sonication times, and the experimental parameters and sample name are shown in the Table 1.

### 2.4. Characterization

The surface morphology of the ACSM composite films was observed using a scanning electron microscope (SEM, Zeiss Sigma300, Oberkochen, German) with an accelerating voltage of 3 kV. The samples were fixed on a metal sample stage with conductive tape and the surface was sprayed with gold. X-ray photoelectron spectra of ACSM composite films were obtained using an X-ray photoelectron spectrometer (XPS, Thermo Scientific K-Alpha, Waltham, MA, USA). Infrared spectra of the films were obtained using a Fourier transform infrared spectrometer (FTIR, IRAffnity-1S, Shimadzu, Japan) in the range of 400–4000 cm^−1^, and the samples were tested using the potassium bromide press method. The thermal stability of the prepared films was tested and analyzed using a simultaneous thermal analyzer (TG, NETZSCH, Bavaria, German) with a sample mass in the range of 6–10 mg, nitrogen as a protective gas, and a temperature increase in the range of 30–700 °C at a rate of 10 °C/min. The thermal properties of the films were evaluated using a differential scanning calorimeter (DSC, TA Q20, Newcastle, DE, USA). Amounts of 6–10 mg of sample were placed in aluminum pans and sealed. Empty capsules were used for reference. The samples were heated between 30–350 °C at a rate of 10 °C/min while exposed to a nitrogen atmosphere at a rate of 10 mL/min. The hydrophilic properties of the ACSM composite films were tested using a water drop angle analyzer (SK-PHb, Chongqing, China). The light transmission of the ACSM composite films was tested using the UV spectrophotometer (JASCO V770, Tokyo, Japan).

### 2.5. Antibacterial Performance Test

The antibacterial performances of ACSM composite membranes were tested using the agar disc diffusion method: 0.1 mL of *E. coli* or *S. aureus* bacterial solution was pipetted onto the solidified agar surface, and the solution was spread evenly with a spreader. After a short period, a circular filter paper (6 mm in diameter) with the membrane was placed on an agar nutrient. CSM composite membranes without AgNO_3_ were used as the blank control group, and CSM composite membranes with the precursor solution containing AgNO_3_ and glycol (ACSM-7) were used as another control group. After 10 min of static standing, the Petri dishes were inverted and cultured in a constant temperature and humidity chamber at T = 37 °C and H = 70% for 24 h. Afterward, the diameter of the inhibition zone around the filter paper was measured using a caliper to evaluate the antibacterial performance of the ACSM composite membranes.

## 3. Results and Discussion

### 3.1. Morphology of ACSM Composite Films

Figure 2 shows the macroscopic images of composite membranes under different parameters. The membrane of the CSM without AgNO_3_ and ultrasound treatment was white, soft, and thin. The color of the ACSM membranes after ultrasonic treatment in a precursor solution was deepened, orange or brown, elastic, and slightly curled at the edges. The change in membrane color was due to the surface plasmon resonance (SPR) of AgNPs, and the slight difference in membrane color at different parameters was due to the content and size of AgNPs [29].

SEM images of as-prepared ACSM composite membranes are shown in Figure 3. At 5K magnification, the CSM membrane has a layered structure of modified carboxymethyl cellulose on the left side of the surface and many long fibers on the right side (Figure 3a). When the magnification is up to 50 K, its surface is a layered structure of modified cellulose (Figure 3b). As shown in Figure 3c–e, the Ag NPs were uniformly and densely distributed on the membrane surface as the sonication time increased from 15 to 45 min. When the time increased to 60 min, Ag NPs were sparsely distributed (Figure 3f). From Figure 3f–h, the Ag NPs on the surface distributed unevenly and easily agglomerated when the reducing agent content was low. When the reducing agent content increased to 10 g, the Ag NPs were distributed tightly and uniformly with less reunion (Figure 3e).

### 3.2. ACSM Composite Membranes Structure and Components

XPS spectra of the ACSM composite membrane are shown in Figure 4. From Figure 4a, the ACSM composite membrane mainly contains carbon, oxygen, and silver elements. As shown in Figure 4b, the characteristic peaks of Ag 3d consisted of the twin peaks of Ag 3d_5/2_ at 368.28 eV and Ag 3d_3/2_ at 374.28 eV; the difference in binding energy between the two characteristic peaks was 6 eV, which confirmed that the elemental Ag existed in the form of elemental silver in the prepared ACSM films. The binding energies of both Ag 3d_5/2_ and Ag 3d_3/2_ of the prepared ACSM membranes were lower than those of standard monolithic silver, which was mainly due to the interaction between Ag NPs and CMC/SR/MBA [30].

As shown in Figure 5a, the characteristic peaks of the CSM membrane at 3309 cm^−1^ are related to the stretching vibration of the -OH group, and at 2360 cm^−1^ are related to the vibration of the -C=O- group. The characteristic peaks of the ACSM membrane at 3379 cm^−1^ are related to the stretching vibration of the -OH group, at 2360 cm^−1^ related to the vibration of the -C=O- group, and at 1604 cm^−1^ related to the asymmetric stretching vibration of the -COO group in carboxylates. The characteristic peak at 2360 cm^−1^ is the vibration of the -C=O- group, the characteristic peak at 1604 cm^−1^ is related to the asymmetric stretching vibration of the -COO group in carboxylate, and the characteristic peak at 1087 cm^−1^ is related to the stretching vibration of the C-O-C group. As shown in Figure 5b, the transmittance at the -OH peak is different, indicating that the number of -OH functional groups in the membrane is different. Higher transmittance indicates a higher number of -OH groups, which enhances the ability to form hydrogen bonds and allows for increased intermolecular interactions. As shown in Figure 5c, both CSM and ACSM membranes showed a characteristic peak at 671 cm^−1^, and the ACSM membrane showed a characteristic peak at 532 cm^−1^, indicating the appearance of C-O-Me bonds.

### 3.3. Thermal Stability and Hydrophobicity of ACSM Composite Membranes

As shown in Figure 6a, the CSM membrane shows two main weight loss stages including 30–150 °C and 150–315 °C, and the weight loss rate is 6.48% and 57.26%. The weight loss for the first stage is mainly due to the evaporation of water in the membranes, while the mass loss for the second stage is due to the degradation of CMC and SR. ACSM membranes also exhibit two main weight loss intervals, including 30–260 °C and 260–350 °C. The weight loss of ACSM-1 to ACSM-6 is shown in Table 2. The weight loss for the first stage of the ACSM membranes is due to the evaporation of water in the membrane, some ethylene glycol attached to the surface of the membrane, and the partial degradation of molecular chains in CMC and SR, while the weight loss for the second stage is due to the degradation of CMC and SR. As shown in Figure 6b, the CSM membrane has two weight loss peaks, and the temperature of the maximum weight loss rate (T_p_) is 270 °C when the weight loss rate is 3.10. ACSM membranes also have two weight loss peaks. The T_p_ and its corresponding mass are shown in Table 2.

According to Table 2, we can find that as the ultrasonic time increased, the T_p_ of the ACSM membranes decreased while the mass loss rate at the T_p_ first decreased and then increased, indicating that the thermal stability of ACSM membranes first decreased and then increased. With the decrease in reducing agent content, the T_p_ of the ACSM membranes gradually decreased, and the mass loss rate at the T_p_ increased, indicating that the thermal stability of the ACSM membranes gradually deteriorated. Comparing the TG and DTG data, ACSM-4 has better thermal stability.

In Figure 6c and Table 3, the CSM and ACSM membranes had two heat absorption peaks. The CSM membrane had a broad heat absorption peak in the 39–203 °C range, reflecting the evaporation of water and other volatile matter from the membrane. It also had a second sharp heat absorption peak at 243–304 °C, reflecting the melting and degradation of the membrane in this temperature range. All ACSM membranes also have a broad heat absorption peak in the 34–159 °C range, reflecting the evaporation of water from the membranes and the volatilization of residual glycol from the surface of the membranes and other volatile matter. In addition, it also had a second sharp heat absorption peak in the 150–280 °C range, reflecting the melting and degradation of the membranes in this temperature range [31]. The peak area of heat absorption indicates the enthalpy change under the peak shape. Compared with the thermal decomposition temperature (T_onset_), the end temperature of thermal decomposition (T_endset_), and the peak area of heat absorption (Δ H) in the DSC curves of ACSM-1 to ACSM-4, the thermal stability of the membranes firstly deteriorated, and then gradually became better with the increasing of the ultrasonic time. According to the DSC curves of ACSM-3, ACSM-5, and ACSM-6, the membrane’s thermal stability gradually worsened with the decreasing content of the reducing agent.

As shown in Figure 7, the water contact angle of the CSM membrane was 42.54°, and the water contact angles of the ACSM membranes were 39.90, 54.43, 59.00, 41.55, 56.44, and 41.81°, respectively. The water contact angle of all membranes was below 90°, which was hydrophilic. As the ultrasonic time increased to 45 min, the water contact angle of the ACSM membranes gradually increased, indicating that the hydrophobicity was improved. However, as the ultrasonic time continued to 60 min, the water contact angle of the ACSM membrane decreased, indicating that the ACSM membrane became hydrophilic. With the decrease in reducing agent content, the water contact angle of the ACSM membrane gradually increased, indicating that the hydrophobicity was improved accordingly. The wettability of a solid surface is determined by the surface chemistry and surface roughness. Low surface free energy and suitable roughness are important factors for the preparation of hydrophobic surfaces. Ag NPs were encapsulated on the membrane surface, and the clusters and small particles formed by the agglomeration of small particles constituted a micro/nanoscale roughness, which reduced its surface free energy and improved its hydrophobic properties [31].

### 3.4. Light Transmission Analysis and Antimicrobial Properties of ACSM Composite Membranes

UV light may adversely affect the nutrients and chemicals (e.g., vitamins, fatty acids, etc.) in certain foods, so membranes with low UV transmittance can effectively block the penetration of UV light, reduce the UV exposure of foods, and help protect their quality and stability. In addition, films with low UV transmittance can reduce oxidation reactions caused by food exposure to UV exposure, thereby delaying spoilage, deterioration, and color change. They help to extend the shelf life of food products and improve their shelf life [32]. The transmittance curves of CSM and ACSM membranes are shown in Figure 8, and the %T values of the samples at different wavelengths in Table 4. The light transmittance of the membranes decreased gradually as the ultrasound time increased, and as the content of the reducing agent increased, the light transmittance of the membranes also decreased. The lowest transmittance was ACSM-3, which indicates that the membrane has a better ability to block UV radiation.

Figure 8b–e shows the data on the circle of inhibition of *E. coli* and *S. aureus* by CSM and ACSM membranes. There was no inhibitory circle around the CSM membrane, and an inhibitory circle appeared around the ACSM membrane after sonication. Using ACSM-7 as a control, the diameter of the inhibitory circle that appeared around it was smaller than the ACSM membrane after sonication. It indicated that Ag NPs played a bacteriostatic role, and its bacteriostatic effect on *E. coli* was slightly better than that of *S. aureus*. The smaller the particle size, the better the bacteriostatic effect of Ag NP-loaded membranes. This is because Ag NP easily adheres to the bacterial cell membrane, enters the cell interior, and accumulates inside it.

As shown in Figure 9, the inhibitory mechanism of Ag NPs is as follows: when Ag NPs are produced and attached to the surface of the bacterial cell membrane, they can interfere with the permeability and respiratory function of the cell; moreover, Ag NPs can enter into the bacteria to disrupt their normal physiological activities and metabolism, and even destroy their bacterial structure, thus inhibiting the bacteria to achieve the antimicrobial effect [33]. In addition, compared with *S. aureus*, ACSM membranes have higher antibacterial activity against *E. coli*. It may be because the outer membrane of *E. coli* is rich in polysaccharides, phospholipids, and proteins, etc. The -COO- and -OH groups in the ACSM membranes can interact with these outer membranes, facilitating the entry of Ag NPs into the outer membrane to destroy *E. coli* cells.

## 4. Conclusions

In this work, we prepared CMC/SR/MBA composite membranes using the casting method, then coated Ag NPs by ultrasonic radiation to prepare Ag NPs/CMC/SR/MBA composite membranes. The Ag NP distribution can be regulated by changing the ultrasonic time and reductant content, achieving their more uniform distribution and dispersion on the ACSM-3 surface. In addition, the ACSM membranes had good thermal stability and an inhibitory effect on *E. coli* and *S. aureus*. Due to the simplicity, eco-friendliness, and industrial applicability of the sonochemical technique, this work offers a new strategy for the fast and efficient preparation of bacteriostatic packaging materials.

## Figures and Tables

**Figure 1 polymers-16-01282-f001:**
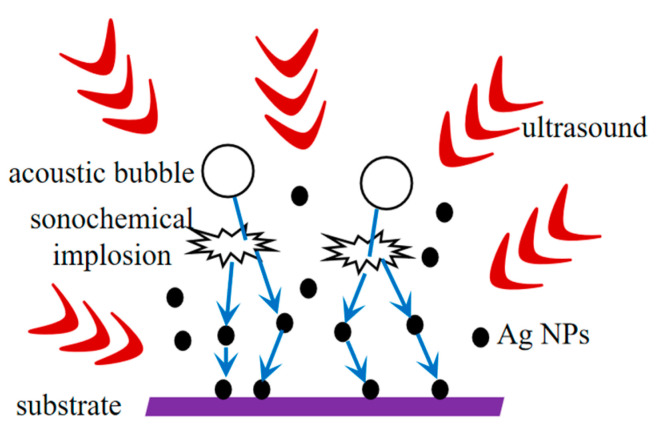
Mechanism diagram of acoustic chemical coating Ag NPs antimicrobial layer.

**Figure 2 polymers-16-01282-f002:**
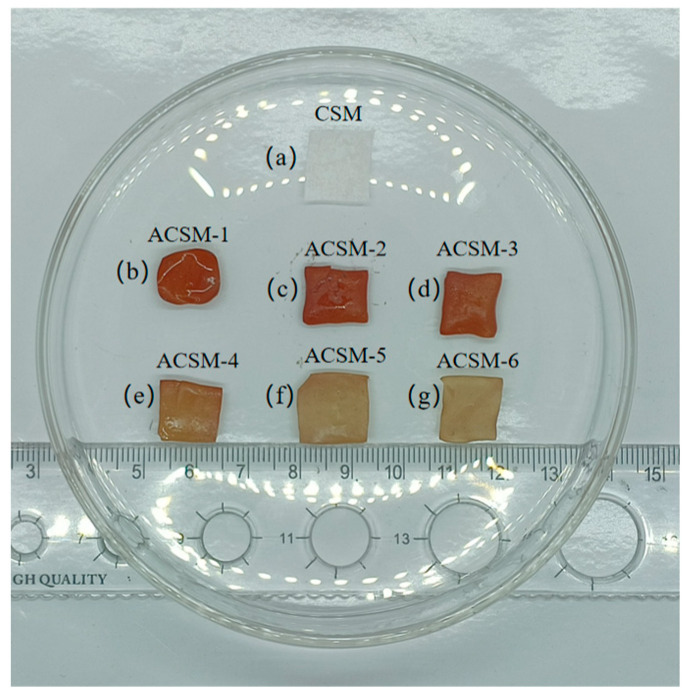
Macroscopic photos of composite membranes under different experimental parameters: (**a**) CSM membrane; (**b**) ACSM-1 (US-15 min, EG-10 g); (**c**) ACSM-2 (US-30 min, EG-10 g); (**d**) ACSM-3 (US-45 min, EG-10 g); (**e**) ACSM-4 (US-60 min, EG-10 g); (**f**) ACSM-5 (US-45 min, EG-8.5 g); (**g**) ACSM-6 (US-45 min, EG-7 g).

**Figure 3 polymers-16-01282-f003:**
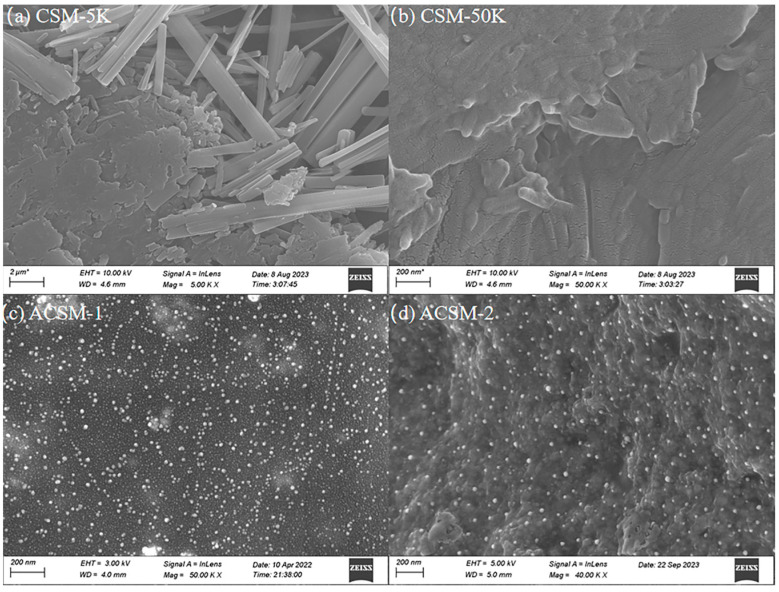
(**a**) SEM image of CSM membrane at 5 K mag; (**b**) SEM image of CSM membrane at 50 K mag; (**c**) SEM of ACSM-1 (US-15 min, EG-10 g); (**d**) SEM image of ACSM-2 (US-30 min, EG-10 g); (**e**) SEM of ACSM-3 (US-45 min, EG-10 g); (**f**) SEM of ACSM-4 (US-60 min, EG-10 g); (**g**) SEM of ACSM-5 (US-45 min, EG-8.5 g); (**h**) SEM of ACSM-6 (US-45 min, EG-7 g).

**Figure 4 polymers-16-01282-f004:**
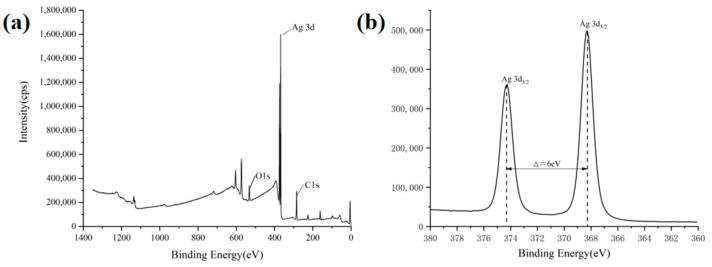
(**a**) Full photoelectron spectra of ACSM composite membranes; (**b**) fine spectrum of Ag.

**Figure 5 polymers-16-01282-f005:**
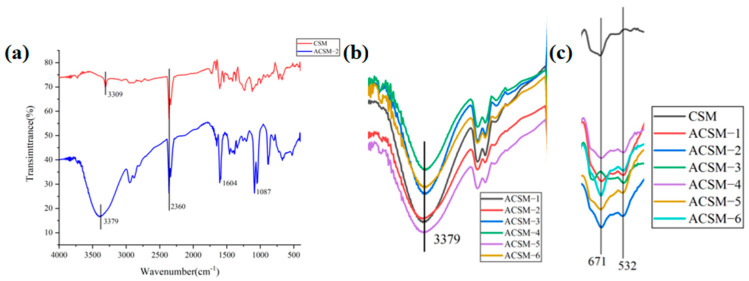
(**a**) FTIR spectra of CSM and ACSM−2 membranes; (**b**) representative images of the peak at 3379 cm^−1^ for the ACSM membranes; (**c**) representative images of the peak at 400–800 cm^−1^ for the CSM membrane and ACSM membranes.

**Figure 6 polymers-16-01282-f006:**
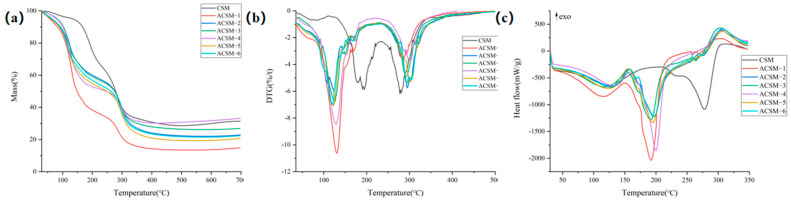
(**a**) TG diagram of CSM membrane and ACSM membranes; (**b**) DTG diagram of CSM membrane and ACSM membranes; (**c**) DSC diagram of CSM membrane and ACSM membranes.

**Figure 7 polymers-16-01282-f007:**
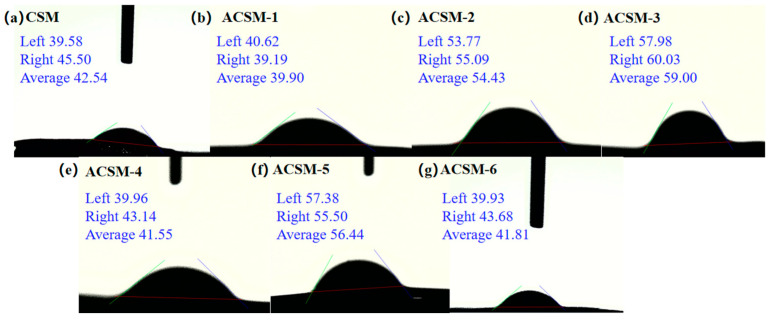
Water contact angle diagram of CSM membrane and ACSM membranes: (**a**) CSM membrane; (**b**) ACSM-1 (US-15 min, EG-10 g); (**c**) ACSM-2 (US-30 min, EG-10 g); (**d**) ACSM-3 (US-45 min, EG-10 g); (**e**) ACSM-4 (US-60 min, EG-10 g); (**f**) ACSM-5 (US-45 min, EG-8.5 g); (**g**) ACSM-6 (US-45 min, EG-7 g).

**Figure 8 polymers-16-01282-f008:**
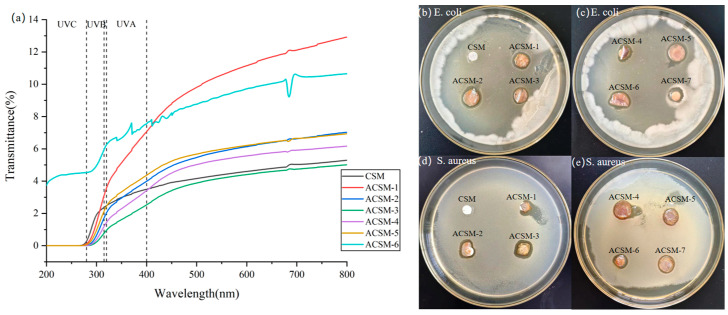
(**a**) Light transmittance of CSM membrane and ACSM membranes; (**b**–**e**) circle of inhibition of prepared ACSM membranes against *E. coli* and *S. aureus*.

**Figure 9 polymers-16-01282-f009:**
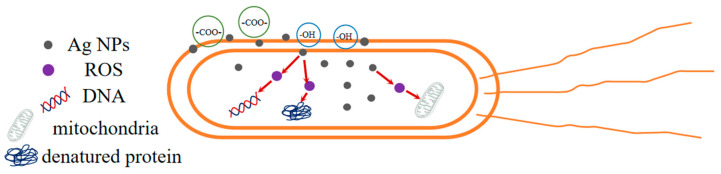
Bacteriostatic mechanism of Ag NPs.

**Table 1 polymers-16-01282-t001:** Sample numbers of ACSM composite membranes and their corresponding experimental parameters.

Sample Number	Reducer EG Content/g	Sonication Time/min
CSM	0	0
ACSM-1	10	15
ACSM-2	10	30
ACSM-3	10	45
ACSM-4	10	60
ACSM-5	8.5	45
ACSM-6	7	45
ACSM-7	10	0

**Table 2 polymers-16-01282-t002:** TGA results for as-prepared ACSM membranes.

Sample Number	Weight Loss between 30–260 °C/%	Weight Loss between 260–350 °C/%	T_p_/°C	Weight Loss Rate at T_p_/(%/t)
ACSM-1	67.76	16.42	130	10.72
ACSM-2	46.40	26.50	124	5.73
ACSM-3	47.06	22.87	125	6.74
ACSM-4	51.06	17.49	129	8.39
ACSM-5	51.52	24.66	121	6.85
ACSM-6	49.60	24.14	120	7.19

**Table 3 polymers-16-01282-t003:** Differential scanning calorimetry data for CSM membrane and ACSM membranes.

Sample Number	Peak 1				Peak 2			
	T_onset_/°C	T_endset_/°C	Δ H/(J/g)	Peak T/°C	T_onset_/°C	T_endset_/°C	Δ H/(J/g)	Peak T/°C
CSM	39.4	203.1	75.6	129	243.3	304.7	36.2	278
ACSM-1	36.7	149.6	70.9	116	150.5	223.4	73.0	192
ACSM-2	34.8	153.3	59.1	130	157.4	261.1	60.5	197
ACSM-3	37.6	152.8	58.0	119	150.9	229.0	53.1	192
ACSM-4	37.2	158.9	53.6	133	158.9	243.5	62.0	200
ACSM-5	37.1	154.2	59.7	124	154.7	279.2	65.4	194
ACSM-6	38.1	158.9	58.4	128	159.3	250.9	55.8	197

**Table 4 polymers-16-01282-t004:** Percentage of light transmission (%T) values of CSM membrane and ACSM membranes.

Sample Number	T at 315 nm/%	T at 400 nm/%	T at 400 nm/%
CSM	2.37	3.49	5.04
ACSM-1	3.10	7.06	12.12
ACSM-2	1.72	3.99	6.63
ACSM-3	0.77	2.53	4.74
ACSM-4	1.19	3.41	5.92
ACSM-5	2.13	4.36	6.65
ACSM-6	5.95	7.55	10.59

## Data Availability

Data are contained within the article.

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
