# Peer review of "Antimicrobial Properties of Carboxymethyl Cellulose/Starch/N’N Methylenebisacrylamide Membranes Endowed by Ultrasound and Their Potential Application in Antimicrobial Packaging"

_polymers, 2024, doi:10.3390/polym16091282_

Round 1

Reviewer 1 Report

Comments and Suggestions for Authors

The authors of the manuscript presented data on a method for producing composites based on carboxymethyl cellulose and silver nanoparticles. Despite the good results, the reviewer did not find the research novelty.

There are a number of comments regarding the data presented in this form.

1. Section Introduction. There is no connection between sentences 2 and 3 in the introductory part. Why mention petroleum-based materials if cellulose and its derivatives are not included and the authors do not consider the removal of such products by materials from cellulose?

2. What is new about the work? The introduction of Ag nanoparticles into any natural or synthetic polymer matrix gives it biocidal properties.

3. Caption for Fig. 2 should be more expanded. You should indicate which samples these are.

4. To Fig. 3. The reviewer did not see in the figure. 3a rod-like fibers. More reminiscent of the layered structure of modified carboxymethyl cellulose. It is more convenient to sign which sample corresponds to what. In this case, the connection between the experimental conditions and the resulting structures will immediately become obvious.

5. To Fig. 5. Did the authors record changes in the IR spectra in the range of 800-400 cm-1 for samples containing Ag? In this area (the appearance of new bands or the shift of existing bands) just indicates the appearance of -O-Me bonds.

6. Why are data for the ACSM-7 sample not presented in tables 2 and 3?

7. Ultimately, what material did the authors want to get? Hydrophilic or hydrophobic? According to the data in Fig. 7, the surface wettability for the ACSM-4 sample is, within the limits of error, the same as for the CSM.

8. Page 9, line 321: Link to Fig. 6. Do TG and DSC diagrams really suggest an inhibitory mechanism?

9. According to the data presented in Table 4, the authors obtained the same results of antibacterial activity (within the error) against gram-positive and gram-negative bacterial cultures. This is a known fact. However, the data on antibacterial effectiveness obtained by the authors do not correlate with the size of Ag nanoparticles. How can this be explained?

By eliminating the above comments, the authors will help improve the level of their article.

Author Response

Thank you very much for taking the time to review this manuscript. The detailed response and corresponding corrections are attached and highlighted in red in the resubmission.

Reviewer 2 Report

Comments and Suggestions for Authors

Dear Authors,

The paper has good structure. The idea of using this composite for antimicrobial packaging is good. I really liked the conclusion. It is concise and well-written. There is a few major issues that should be addressed to.

1) Even you mention everywhere that this is carbomethyl cellulose you actually have in composite also a starch and methylenebisacrylamide which you cannot ignore because it is in the same amount in the composite. Need to address it through hole paper.

2) The other major thing is particle size of AgNPs. How did you calculate it? The SEM image of ACSM-2 is a little bit strange. It would make more sense if the size were similar to ACSM-1 and ACSM-3. Can you check it? In the image it looks like the AgNPs is more then 30 nm if we compare it to 28 nm size of ACSM-1. It is completely different.

3) At almost the all figures you noted that figure form (b) to (g) are ACSMs but you have 6 samples there and in the table 1 you had 7 samples. It would be necessary, in my opinion that you do that note more widely and precisely like, (b) ACSM-1, (c) ACSM-2 etc.

I also found a few minor issues which I will describe in specific comments.

Author Response

(The authors gave the same response as above.)

Reviewer 3 Report

Comments and Suggestions for Authors

The manuscript titled " Antimicrobial properties of carboxymethyl cellulose composite membranes endowed by ultrasound and their potential application in antimicrobial packaging” (polymers-2772212) described the preparation of carboxymethyl cellulose based films substrates, coated with Ag nanoparticles, for food packaging purposes. 

The introduction and state of the art give a quick overview of the topic, but, unfortunately, they fail to show the novelty of the work. Indeed, a quick search using carboxymethyl cellulose and Ag nanoparticles generates a wide list of published papers where they explored these materials for similar purposes. Therefore, my first question is why is this work different from all the rest? By reading the paper, I am not able to answer that. Please revise this. 

For this reason, at this point, the paper fails to capture the attention of the reader. 

The experimental design is ok, but I believe that, in order to increase the soundness of the paper, more tests should be included.  

The results section is not well presented. Sub-section should begin with an image (line 157). This shows lack of care and is not how a scientific paper should be presented. Then SEM analysis begins with a huge mistake (line 169): Ag NPs are not coated; they are coating the membrane. This mistake completely changes the meaning of the sentence.  

Figure 3 is confusing – the name of the membranes should be on it, otherwise it is very difficult to correctly understand to which membrane does a part of the image stand. In addition, the SEM section is poorly written and needs deep revision. It is confusing and most of the sentences fail to make sense. The same happens in XPS and FTIR sections – the text is quite confusing. 

Authors claim that the color change of the membranes was due to the possible oxidation of the Ag NPs. So, what are the implications of the oxidation of the NPs in the membrane characterization and antimicrobial performance? Please clarify it. 

Why not include some inhibitory halo’s figures? This would strongly help the reader to understand the text.  

The conclusions section is incomplete, in my opinion. In addition, authors make claims that are not supported by the presented results. For that reason, I advise the authors to include more characterization work in order to be able to infer that theses membranes could be used for food packaging. 

Overall, I believe that the manuscript should be rethought, rewritten and only then, resubmitted. In its current state I do not think this paper has the quality required to be published in a Q1 journal as “Polymers”. In my opinion, there are some serious flaws in the experimental design, and for that reason, results do not entirely support the claims made in the conclusions. In addition, the novelty of the work is not clear. 

Nonetheless I believe that this manuscript should be published in the journal "Polymers" after minor revisions. 

Extra revisions: 

1. Double spacing or no space at all can be found in some parts of the manuscript. Please improve that. 

2. Line 287: expression “relatively hydrophilic”. This is not a scientific term. Please rewrite this sentence. 

3. Lines 341-342: This sentence repeats itself. Please revise it. 

4. Lines 342-344: This claim is not supported by the presented results. I don’t think that the authors should state this without further investigation.

Comments on the Quality of English Language

Overall, the writing across the entire manuscript is quite poor, and would strongly benefit from a revision. I recommend having a 3rd party (English native) read and edit the manuscript.

Author Response

(The authors gave the same response as above.)

Round 2

Reviewer 1 Report

Comments and Suggestions for Authors

The authors took into account the reviewer's comments. A more correct description of the results obtained increased the level of presentation of the material. Thank you.

Reviewer 3 Report

Comments and Suggestions for Authors

The authors were able to fully answer all my questions. In my opinion, the improvements made are enough to publish the paper.

Comments on the Quality of English Language

English was improved and now the manuscript is much more understandable.